# Towards Variational Generation of Small Graphs

**Martin Simonovsky & Nikos Komodakis**
Université Paris Est, École des Ponts ParisTech
Champs sur Marne, France
`{martin.simonovsky,nikos.komodakis}@enpc.fr`

## Abstract

In this paper we propose a generative model for graphs formulated as a variational autoencoder. We sidestep hurdles associated with linearization of graphs by having the decoder output a probabilistic fully-connected graph of a predefined maximum size directly at once. We evaluate on the challenging task of molecule generation.

## 1 Introduction

Deep learning on graphs has very recently become a popular research topic (Bronstein et al., 2017). Past work has concentrated on learning graph embedding tasks so far, *i.e.* encoding an input graph into a vector representation. This is in stark contrast with fast-paced advances in generative models for images and text. Hence, it is an intriguing question how one can transfer this progress to the domain of graphs, *i.e.* their decoding from a vector representation.

However, learning to generate graphs is a difficult problem. Unlike sequence generation, graphs can have arbitrary connectivity and there is no clear best way how to linearize their construction in a sequence of steps: Vinyals et al. (2015) empirically found out that the linearization order matters when learning on sets. On the other hand, iterative construction of discrete structures during training without step-wise supervision involves discrete decisions, which are not differentiable and therefore problematic for back-propagation.

In this work, we propose to sidestep these hurdles by having the decoder output a probabilistic fully-connected graph of a predefined maximum size directly at once. In a probabilistic graph, the existence of nodes and edges, as well as their attributes, are modeled as independent random variables. The method, coined GraphVAE, is formulated in the framework of variational autoencoders Kingma & Welling (2013) and demonstrated on the task of molecule generation.

**Related Work.** Johnson (2017) constructs a probabilistic (multi)graph according to a sequence of input sentences to answer a query. While our model also outputs a probabilistic graph, we do not assume having a prescribed order of construction transformations available and we formulate the learning problem as an autoencoder. Xu et al. (2017) learns to produce a scene graph from an input image and a set of object proposals. In contrast, our method does not need to specify the number of nodes or the structure explicitly. Related work pre-dating deep learning includes random graphs (Erdos & Rényi, 1960; Barabási & Albert, 1999) or stochastic blockmodels (Snijders & Nowicki, 1997). Cheminformatics has exploited progress made in text generation for string representation of molecules (Gómez-Bombarelli et al., 2016; Olivecrona et al., 2017; Segler et al., 2017). As the syntax is brittle, many invalid strings tend to be generated, which has been recently addressed by Kusner et al. (2017) by incorporating grammar rules into decoding. While encouraging, their approach does not guarantee semantic (chemical) validity, similarly as our method. An advantage of a graph representation to text is the possibility to predict attributes in addition to the base structure.

## 2 Method

Our main idea is to output a probabilistic fully-connected graph and use a standard graph matching algorithm to align it to the ground truth. We observe the task can become much simpler if we restrict

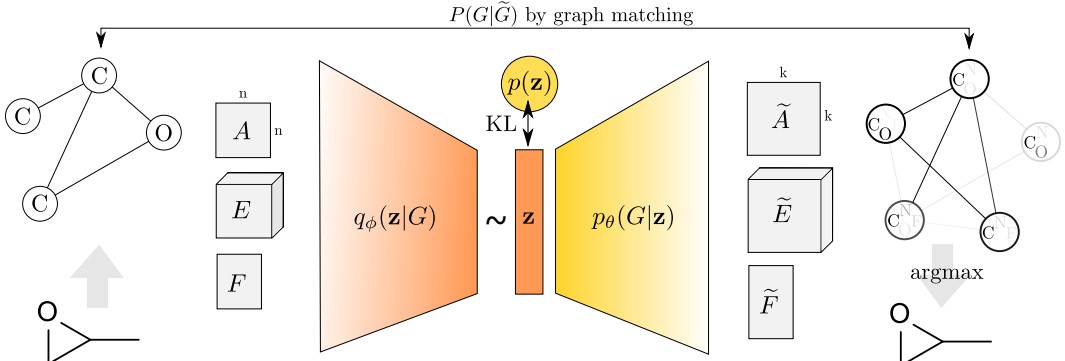

Figure 1: Illustration of the proposed variational graph autoencoder. Starting from a discrete attributed graph $G = (A, E, F)$ on $n$ nodes (*e.g.* a representation of propylene oxide), stochastic graph encoder $q_\phi(\mathbf{z}|G)$ embeds the graph into continuous representation $\mathbf{z}$. Given a point in the latent space, our novel graph decoder $p_\theta(G|\mathbf{z})$ outputs a probabilistic fully-connected graph $\widetilde{G} = (\widetilde{A}, \widetilde{E}, \widetilde{F})$ on predefined $k \geq n$ nodes, from which discrete samples may be drawn. Reconstruction ability of the autoencoder is facilitated by approximate graph matching for aligning $G$ with $\widetilde{G}$.

the domain to the set of all graphs on maximum $k$ nodes, where $k$ is fairly small (tens). Under this assumption, handling dense graph representations is still computationally tractable.

**Graph Decoder.** We propose to make the decoder output a probabilistic fully-connected graph $\widetilde{G} = (\widetilde{A}, \widetilde{E}, \widetilde{F})$ on $k$ nodes at once. The predicted adjacency matrix $\widetilde{A} \in [0, 1]^{k \times k}$ contains both node probabilities $\widetilde{A}_{a,a}$ and edge probabilities $\widetilde{A}_{a,b}$ for nodes $a \neq b$. The edge attribute tensor $\widetilde{E} \in \mathbb{R}^{k \times k \times d_e}$ indicates class probabilities for edges and, similarly, the node attribute matrix $\widetilde{F} \in \mathbb{R}^{k \times d_n}$ contains class probabilities for nodes. At test time, a point estimate of $\widetilde{G}$ can be obtained by taking argmax in $\widetilde{A}, \widetilde{E}$, and $\widetilde{F}$, which can result in a discrete graph on less than $k$ nodes.

**Objective Function.** Let $G = (A, E, F)$ be a graph on $n \leq k$ nodes. We wish to learn an encoder and a decoder to map between the space of graphs $G$ and their continuous embedding $\mathbf{z} \in \mathbb{R}^c$, see Figure 1. The whole model is formulated as a variational autoencoder Kingma & Welling (2013), trained by minimizing $\mathcal{L}(\phi, \theta; G) = \mathbb{E}_{q_\phi(\mathbf{z}|G)}[-\log p_\theta(G|\mathbf{z})] + \mathrm{KL}[q_\phi(\mathbf{z}|G)||p(\mathbf{z})]$. The reconstruction likelihood $p_\theta(G|\mathbf{z}) = P(G|\widetilde{G})$ enforces high similarity of sampled generated graphs to the input graph $G$. We use a simplistic isotropic Gaussian prior $p(\mathbf{z}) = N(0, I)$ for regularization.

Since no particular ordering of nodes is imposed in either $\widetilde{G}$ or $G$ and matrix representation of graphs is not invariant to permutations of nodes, we assume knowledge of a binary assignment matrix $X \in \{0, 1\}^{k \times n}$, defined below, to map information between both graphs. Specifically, input adjacency matrix is mapped as $A' = XAX^T$, whereas the predicted node attribute matrix and slices of edge attribute matrix are transferred as $\widetilde{F}' = X^T \widetilde{F}$ and $\widetilde{E}'_{\cdot,\cdot,l} = X^T \widetilde{E}_{\cdot,\cdot,l} X$. The maximum likelihood estimate for adjacency is the cross-entropy $\log p(A'|\mathbf{z}) = 1/k \sum_a A'_{a,a} \log \widetilde{A}_{a,a} + (1 - A'_{a,a}) \log(1 - \widetilde{A}_{a,a}) + 1/k(k-1) \sum_{a \neq b} A'_{a,b} \log \widetilde{A}_{a,b} + (1 - A'_{a,b}) \log(1 - \widetilde{A}_{a,b})$, for node attributes $\log p(F|\mathbf{z}) = 1/n \sum_i \log F_{i,\cdot}^T \widetilde{F}'_{i,\cdot}$, and for edge attributes $\log p(E|\mathbf{z}) = 1/(||A||_1 - n) \sum_{i \neq j} \log E_{i,j,\cdot}^T \widetilde{E}'_{i,j,\cdot}$. The formulation considers existence of both matched and unmatched nodes and edges but attributes of only the matched ones. The overall reconstruction loss is a weighed sum of the previous terms: $-\log p(G|\mathbf{z}) = -\lambda_A \log p(A'|\mathbf{z}) - \lambda_F \log p(F|\mathbf{z}) - \lambda_E \log p(E|\mathbf{z})$.

**Graph Matching.** The goal of graph matching is to find correspondences $X \in \{0, 1\}^{k \times n}$ between nodes of graphs $G$ and $\widetilde{G}$ based on the similarities of their node pairs $i, j \in G$ and $a, b \in \widetilde{G}$ defined as $S((i, j), (a, b)) = (E_{i,j,\cdot}^T \widetilde{E}_{a,b,\cdot}) A_{i,j} \widetilde{A}_{a,b} \widetilde{A}_{a,a} \widetilde{A}_{b,b}[i \neq j \wedge a \neq b] + (F_{i,\cdot}^T \widetilde{F}_{a,\cdot}) \widetilde{A}_{a,a}[i = j \wedge a = b]$. The first term evaluates similarity between edge pairs and the second term between node pairs.

Table 1: Performance of QM9 models over varied embedding size $c$. Baselines listed only for the embedding size $c$ with the highest Valid ratio.

|  | $\log p_\theta(G|\mathbf{z})$ | Valid | Unique | Novel |
|---|---|---|---|---|
| Ours $c = 20$ | -0.660 | 0.485 | 0.457 | 0.575 |
| Ours $c = 40$ | -0.537 | 0.542 | 0.618 | 0.617 |
| Ours $c = 60$ | -0.486 | 0.517 | 0.695 | 0.570 |
| Ours $c = 80$ | -0.482 | 0.557 | 0.760 | 0.616 |
| CVAE $c = 60$ | – | 0.103 | 0.675 | 0.900 |
| GVAE $c = 20$ | – | 0.602 | 0.093 | 0.809 |

Figure 2: Decodings over a random 2D plane in **z**-space. Chemically invalid graphs in red.

We use max-pooling matching by Cho et al. (2014), a simple but effective algorithm amendable to batch mode, for obtaining a continuous assignment matrix $X^*$, which we discretize as $X$ using Hungarian algorithm to obtain a strict one-on-one mapping. While this operation is non-differentiable, gradient can still flow to the decoder directly through the loss function and training convergence proceeds without problems. To summarize, our method aims to find the best graph matching and then further improve on it by gradient descent on the loss.

**Details.** A feed forward network with edge-conditioned graph convolutions (Simonovsky & Komodakis, 2017) is used as encoder with the graph-level output model of Li et al. (2015b). The decoder is a deterministic multi-layer perceptron with three outputs under sigmoid or softmax activations in its last layer. The proposed model is expected to be useful only for generating small graphs due to growth of number of parameters ($O(k^2)$) and matching complexity ($O(k^4)$). Nevertheless, for many applications even generation of small graphs is still very useful.

## 3 EVALUATION

Graph representation of molecules is a convenient testbed for generative models due to canonical visualization and automated chemical validation of samples. Chemical constraints on compatible types of bonds and atom valences make the space of valid graphs complicated and molecule generation challenging. In fact, a single addition or removal of edge or change in atom or bond type can make a molecule chemically invalid. We compare our model to the character-based generator of Gómez-Bombarelli et al. (2016) (CVAE) and the grammar-based generator of Kusner et al. (2017) (GVAE) on QM9 dataset (Ramakrishnan et al., 2014) of about 134k organic molecules of up to 9 heavy atoms.

**Quantitative Evaluation.** The quality of a decoder can be evaluated by the validity and variety of generated graphs. We draw $n_s = 10^4$ samples $\mathbf{z}^{(s)} \sim p(\mathbf{z})$ and compute the discrete point estimate of their decodings $\hat{G}^{(s)} = \arg \max p_\theta(G|\mathbf{z}^{(s)})$. Let $V$ be the list of chemically valid samples from $\hat{G}^{(s)}$. We are interested in the ratio Valid $= |V|/n_s$, the fraction of unique correct graphs Unique $= |\text{set}(V)|/|V|$, and the fraction of novel graphs Novel $= 1 - |\text{set}(V) \cap \text{QM9}|/|\text{set}(V)|$.

In Table 1, up to 55% of generated molecules are chemically valid. It is also remarkable that about 60% of generated molecules are out of the dataset, *i.e.* the network has never seen them during training. Looking at the baselines, CVAE can output only very few valid samples as expected, while GVAE generates the highest number of valid samples (60%) but of very low variance (less than 10%). We observe reconstruction loss decrease due to larger $c$ providing more freedom up to some level.

**Qualitative Evaluation.** To visually judge the quality and smoothness of the learned embedding **z**, we decode points sampled along a random 2D plane in Figure 2 (for $c = 40$ and within 5 units from center of coordinates). The image shows a varied and fairly smooth mix of molecules.

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
