# OpenReview forum: "Towards Variational Generation of Small Graphs"
_ICLR.cc/2018/Workshop — Accept_

### Official Review · AnonReviewer3 · 2018-03-09
**A generative model of small graphs with a deep autoencoder**

**Rating:** 7
**Confidence:** 3

**Review:**

The paper considers the problem of learning to generate small graphs. The authors use an autoencoder. The difficult part is the decoder, i.e. generating a graph from a vector. The output of the decoder is a probabilistic fully connected graph. The authors define a loss based on a matching between the ouput probabilistic graph and the input graph. The paper is clear, the problem is significant and difficult.
Pros
* significant problem of generating graphs with deep networks
* clear limit of the work to small graphs because of complexity issues
Cons
* The paper should give some precisions on the experimental process. For instance k and n should be given
* The generation of graphs is not clear to me. The ouput of the decoder is a probabilistic fully connected graph. I do not see how the graphs are generated. The number of valid graphs seems very optimistic to me when generating graphs from such a probabilistic model.

---

### Official Review · AnonReviewer2 · 2018-03-09
**An interesting paper worth to be presented.**

**Rating:** 7
**Confidence:** 2

**Review:**

This paper provides a generative model of graphs with the variational auto-encoder and graph matching algorithm. Although the proposed model cannot scale up to graphs with a large number of nodes, overall I think the suggested method is worth to be presented in the workshop.

Needs to be cleared in the text:
- Argmax operation on predicted adjacency matrix A: It is not clear how to perform argmax on the adjacency matrix, unlike edge and node attribute matrices. Did you keep nodes and edges over a certain threshold in probability?
- Symmetricity on the adjacency matrix. The predicted graphs are the directed graph whereas the molecular graphs are undirected.
- Structures of encoder and decoder network. Any reference to the network structures?

Suggestions on evaluation metrics:
The proposed three evaluation metrics are quite interesting and shows some properties of compared methods, but they also have some limitations as well. For example, it would be more meaningful to measure novelty given the same number of unique samples generated from different models. To do this, it will be also interesting to show that how many samples needed to be drawn to obtain a certain amount of unique samples. This metric will be more valuable to practitioners in chemistry. You may find some models cannot generate unique samples more than some threshold at the end. Valid score on the same number of unique samples would be also interesting as well.

---

### Decision · Program_Chairs · 2018-03-20
**ICLR 2018 Workshop Acceptance Decision**

**Decision:**

Accept

**Comment:**

Congratulations, your paper was accepted to the ICLR workshop.